# Does the presence of general practitioners in emergency departments affect quality and safety in English NHS hospitals? A retrospective observational study

James Gaughan [1], Dan Liu,[1] Nils Gutacker,[1] Karen Bloor [2], Tim Doran,[2] Jonathan Richard Benger[3]

¹Centre for Health Economics, University of York, York, UK
²Department of Health Sciences, University of York, York, UK
³Faculty of Health and Applied Sciences, University of the West of England, Bristol, UK

**Correspondence to**
Dr James Gaughan;
james.gaughan@york.ac.uk

## ABSTRACT

**Objectives** Emergency departments (EDs) in NHS hospitals in England have faced considerable increases in demand over recent years. Most hospitals have developed general practitioner services in emergency departments (GPEDs) to treat non-emergency patients, aiming to relieve pressure on other staff and to improve ED efficiency and patient experience. We measured the impact of GPED services on patient flows, health outcomes and ED workload.

**Design** Retrospective observational study. Differences in GPED service availability across EDs and time of day were used to identify the causal effect of GPED, as patients attending the ED at the same hour of the day are quasi-randomly assigned to treatment or control groups based on their local ED's service availability.

**Participants** Attendances to 40 EDs in English NHS hospitals from April 2018 to March 2019, 4 441 349 observations.

**Primary and secondary outcomes measured** Outcomes measured were volume of attendances, 'non-urgent' attendances, waiting times over 4 hours, patients leaving without being treated, unplanned reattendances within 7 days, inpatient admissions and 30-day mortality.

**Results** We found a small, statistically significant reduction in unplanned reattendances within 7 days (OR 0.968, 95% CI 0.948 to 0.989), equivalent to 302 fewer reattendances per year for the average ED. The clinical impact of this was judged to be negligible. There was no detectable impact on any other outcome measure.

**Conclusions** We found no adverse effects on patient outcomes; neither did we find any evidence of the hypothesised benefits of placing GPs in emergency settings beyond a marginal reduction in reattendances that was not considered clinically significant.

## INTRODUCTION

Emergency department (ED) crowding has been described as a 'worldwide public health problem'[1] and even an 'international crisis'.[2] Crowding, which has been linked with various measures of quality of care,[3] can result from the volume of patients attending (input), delays in the ED (throughput) or blocks to patients leaving (output),[4] for example, problems with accessing hospital beds or social care support.

In common with many healthcare systems, EDs in the English NHS have faced considerable growth in demand over recent years.[5] Efforts to reduce unnecessary attendances to EDs (input), such as the introduction of walk-in primary care centres and NHS telephone and online advice services, have been shown to address unmet needs but have not changed trends in ED attendance.[6] Policy attention has consequently switched to throughput of patients inside the ED. A substantial minority of attendances are suitable for treatment by a primary care physician—known in the UK as a general practitioner (GP); Mason *et al*[7] estimated this proportion to be around 23% of adults and 31% of children attending EDs in England. Following the recommendations of a national review of emergency services,[8] most acute care NHS hospitals have developed general practitioner services in emergency departments (GPEDs) to treat non-urgent patients, aiming to relieve pressure on other ED staff and to improve ED efficiency and patient experience.[9 10] Different models of GPs working in or alongside the ED have emerged; some are fully integrated within the

### Strengths and limitations of this study

► Large sample size (all emergency department attendances at 40 English hospitals over 1 year).
► Time of patient arrival used to support a causal interpretation.
► Wide range of performance measures analysed.
► Inability to identify the healthcare professional seen by patients prevented direct comparison.
► Data analysis is based on a number of assumptions.

ED, others 'stream' patients either to GPs working alongside EDs or to outside services, which may be offered on site or off site.[11]

Evidence of the effectiveness of GP or other primary care practitioners in EDs is limited to date. A 2018 Cochrane review found only four studies (one randomised trial and three non-randomised studies) that evaluated the effects of introducing GPs or emergency nurse practitioners.[12] These studies did not examine safety and had inconsistent results, leading reviewers to comment that the evidence was insufficient to draw conclusions for practice or policy. The Cochrane review has relatively stringent methodological inclusion criteria (randomised trials, interrupted time series, and controlled before and after studies only), but more inclusive reviews have similarly found inadequate evidence on the effectiveness of this approach.[13 14] A narrative review including all study types from seven countries (Netherlands, England, Australia, Ireland, Spain, Sweden and Switzerland)[13] described a paradoxical increase in attendances, which the authors attributed to provider-induced demand, and they concluded that any marginal savings per patient are likely to be overshadowed by the overall cost of the new service. A realist review including studies from the UK, six European Union countries, Australia, USA, Canada, Singapore and New Zealand[14] found that GPEDs may shorten process times for non-urgent patients, but that there is little evidence that this frees up ED staff time; this study also raised potential concerns about provider-induced demand.

As part of a mixed methods study to explore the impacts of GPs working in or alongside the ED,[15] we analysed data from hospitals in England with a well-established and clearly defined GPED service during the financial year of April 2018–March 2019, using the presence of a GP service as the intervention. The study aim was to measure the impact of GPED services on patient throughput in the ED (including waiting time, treatment in the ED and unplanned reattendance). Given the association between ED crowding and the quality and safety of patient care,[3] we also investigated patient health outcomes, including 30-day mortality.

## METHODS
### Study design and data sources
We undertook a retrospective observational study of all clinical activity in type 1 EDs (ie, EDs that are open 24 hours a day and led by a consultant in emergency medicine) in English NHS hospital trusts during the period of 1 April 2018–31 March 2019. Complete data on availability and type of GPED service, derived from a national mapping survey,[16] were available for 40 of England's 137 type 1 EDs. The unit of analysis was an individual ED attendance.

The primary data source was the Hospital Episode Statistics Accident & Emergency (HES A&E) dataset.[17] HES A&E is an administrative dataset which reports information about all attendances to all hospital EDs in England and forms the basis for hospital reimbursement. The dataset provides information on patients' sociodemographic characteristics (age (in years), sex and local area deprivation profile based on the Index of Multiple Deprivation (IMD) 2019.[18] There were 4 610 364 attendances to the 40 EDs for which we observed GPED availability times. Of these, we excluded 116 316 (2.5%) observations with missing deprivation score due to unknown patient place of residence, 333 (0.01%) observations with missing sex and 52 366 (1.14%) observations with missing age. This left a main sample of 4 441 349 observations. The timing of patient attendance and discharge from ED (recorded as date, hour and minute) and the discharge destination are also included within the dataset. Full technical details of variables within the HES dataset can be found online (https://digital.nhs.uk/data-and-information/data-tools-and-services/data-services/hospital-episode-statistics/hospital-episode-statistics-data-dictionary).

Information on the date of death for deceased patients was provided by the Office for National Statistics and linked to HES A&E by NHS Digital.[19]

HES A&E data do not identify the treating healthcare professional, and it was therefore not possible to ascertain whether a patient was seen by a GP or another member of staff. Hospital-reported GPED opening hours were used instead to assign ED attendances to treatment and control groups based on the potential exposure to GPED services at the patient's time of arrival to the ED. Data on opening and closing hours of GPED services for individual hospitals by day of the week were obtained via the mapping survey.[16] EDs were also asked to identify their local model of GPED service, which was prespecified to be either (1) integrated, with the GP working as part of the ED team; (2) parallel, where the GP service is separate from but located adjacent to the ED; or (3) off-site, with the GP service provided at a separate site on the same hospital campus. Only hospital EDs that provided complete data as part of this national survey were included in the analyses.

### Outcome measures
We investigated the impact of GPED on a range of different measures of ED performance and patient outcomes (table 1). All outcome measures were defined at individual patient level except volume of activity, which is measured at ED–hour-day level. Outcomes could not be calculated for all attendances due to missing information recorded in the HES dataset. For example, discharge destination was not recorded for 14% of patients so that outcomes 3 ('untreated') and 5 ('admitted to ward') could not be calculated for these patients. For outcomes 4 ('non-urgent attendance') and 6 ('30-day mortality'), missingness could not be ascertained because information about the outcome is defined as the presence of some recorded information. If this information is absent, it was assumed that the outcome had not occurred. Finally, attendances occurring within the final 7 (30) days

**Table 1**  Outcome measures and summary statistics for all EDS in England and those included in the analysis

| ID | Outcome | Definition | Unit of analysis | Included EDs (n=40) | | | | Excluded EDs (n=97) | | | | Missing (%) (included EDs only) |
|---|---|---|---|---|---|---|---|---|---|---|---|---|
| | | | | Proportion of patients (%) | Observations (millions) | Mean (SD) | Observations (000's) | Proportion of patients (%) | Observations (millions) | Mean (SD) | Observations (000's) | |
| **Patient flow and outcomes** | | | | | | | | | | | | |
| 1 | Wait over 4 hours | Time from admission to discharge was more than 4 hours. | Patient | 16.49 | 4.4 | | | 19.77 | 11.6 | | | 0.6 |
| 2 | Unplanned reattendance | Patient made an unplanned reattendance to the same or different ED within 7 days of discharge. | Patient | 9.01 | 4.2 | | | 8.67 | 11.1 | | | 3.5 |
| 3 | Untreated | Patient left ED without being treated. | Patient | 2.36 | 3.8 | | | 2.13 | 10.8 | | | 14.2 |
| 4 | Non-urgent attendance[7 23] | An attendance is 'unnecessary' or 'non-urgent' if *all* of the following conditions are met:<br>▶ No investigations were recorded or they were limited to urinalysis, pregnancy test or dental investigation.<br>▶ No treatments were recorded or were limited to guidance, written advice, verbal advice, recording of vital signs, dental treatment or prescription of medicines.<br>▶ Treatment or follow-up was limited to primary care or patient left before being treated.<br>▶ Attendance was not by ambulance. | Patient | 11.50 | 4.4 | | | 10.69 | 11.6 | | | Unknown |
| 5 | 30-day mortality | Date of death of the patient is within 30 days after ED attendance. | Patient | 1.58 | 4.1 | | | 1.74 | 10.6 | | | Unknown |
| 6 | Admission to ward | Patient admitted as an inpatient following ED attendance | Patient | 23.65 | 3.8 | | | 23.52 | 10.8 | | | 14.2 |
| **ED workload** | | | | | | | | | | | | |
| 7 | Volume of attendances | Count of attendances per hour of day and day of week | Provider–hour-day | | | 12.68 (9.13) | 350 | | | 13.37 (11.59) | 850 | 0.0 |

ED, emergency department.

of the study period were not included when analysing unplanned reattendances (30-day mortality) as reattendance or death might have occurred within the relevant time frame but outside the study period. Information about the specific HES A&E variables, the values used to identify each outcome and the reasons for missingness in each outcome measure, are presented in online supplemental appendix A. Observations with missing outcome information were only excluded from the statistical analysis of the relevant outcome.

## Statistical analysis

The statistical analysis relies on differences in GPED service availability across EDs and time of day to identify the causal effect of GPED on outcomes. Hospitals operate different opening hours for their GPED services and few offer 24/7 coverage. For example, a service might begin at 08:00 and end at 23:00 in one hospital but run from 24:00 to 20:00 in another hospital. As such, at a given hour, patients attending some EDs will have access to GPED services, whereas patients attending some other EDs will not. As patients are likely to attend the ED nearest to their usual place of residence and are unlikely to plan the timing of their attendance, this created a natural experiment in which patients attending the ED at the same hour of the day are quasi-randomly assigned to treatment or control groups based on their local GPED service availability. To estimate the impact of GPED availability, we considered all 24 hours of the day over the full study period from 1 April 2018 to 31 March 2019.

Logit and Poisson regression models were estimated to identify the effect of treatment status on outcomes at patient and provider levels. Each outcome was analysed separately. All models controlled for hospital and hour-by-day of week fixed effects, month-of-year fixed effects and, in the case of the attendance-level regressions, potential residual confounders in the form of patient age (coded as 0, 1, 2, 3, 4, 5, 6–10, 11–15 … 91–95, 96+), sex, age–sex interactions, an indicator for arrival by ambulance and area-level socioeconomic deprivation based on place of residence (IMD) score grouped in equal quintiles. The hospital fixed effects capture time-invariant differences in performance across EDs that reflect hospital-specific factors such as management quality, building infrastructure, and the quality and availability of substitute healthcare services within the local health economy.[20] The hour-by-day fixed effects capture differences in service availability and patient acuity over the course of the day that follow a common pattern across all hospitals in England. Attendances in the last 7 or 30 days of the sample period were excluded from analysis for outcomes 7-day reattendance and 30-day mortality, respectively.

Our main analyses assumed a common effect across GPED models. Secondary analyses allowed for subgroup effects by type of GPED model.

The estimation of the impact of GPED was limited to time periods where there was variation in GPED service availability across hospitals. While all attendances to analysed EDs in the study period were analysed, patients attending EDs during hours of the day when all/none of the EDs operated a GPED service contributed to the identification of hospital fixed effects and the effect of observed confounders but did not contribute to the statistical identification of the effect of GPED services on outcomes. This group includes 257 532 attendances (5.8% of the full sample).

Estimates are reported as ORs or incidence rate ratios with associated 95% CIs. Figures are rounded to the third decimal due to the large number of ED attendances. Bonferroni-Dunn correction was applied to adjust CIs for multiple testing while ensuring an overall family-wise error rate of 5%.[21] SEs are clustered at ED level. All analyses were performed in Stata V.16.

## Patient and public involvement

We held meetings with a patient and public research advisory group throughout the study. We discussed the study methods with this group and presented results for feedback at various stages, modifying the research and its interpretation in response to their feedback. Our final results were disseminated to the research advisory group and others through a collaborative workshop. Our study was an analysis of secondary data; therefore, we did not recruit patients as research participants.

## RESULTS
### GPED services

Data on GPED service opening hours and GPED model were available for 40 EDs, with a total of 4.6 m attendances between April 2018 and March 2019. Most EDs had a GP present between 10:00 and 22:00 on all days of the week (figure 1), which coincides with the time of day in which workload tends to be highest (figure 2). Some hospitals operate different GPED service opening hours during weekdays and weekends, with two hospitals offering no GPED services on weekends. One or more GPs were present in EDs in our sample for a median of 13 hours (IQR 11–15) on weekdays and 13 hours (IQR 9–15) on weekends.

Demographic characteristics of patients attending EDs where a GPED service was available and where GPED was not available were similar, on average, and followed a common pattern over the course of the day (figure 3). The only exception is deprivation, with patients attending EDs where a GPED service was available during the early hours of the day being, on average, from less deprived neighbourhoods.

### Impact of GPED

The results of the main analysis are presented in table 2. There was no statistically significant effect of GPED on any of the outcome measures aside from a small, statistically significant reduction in unplanned reattendances within 7 days (OR 0.968, 95% CI (unadjusted) 0.948 to 0.989). This effect was only detectable for integrated and

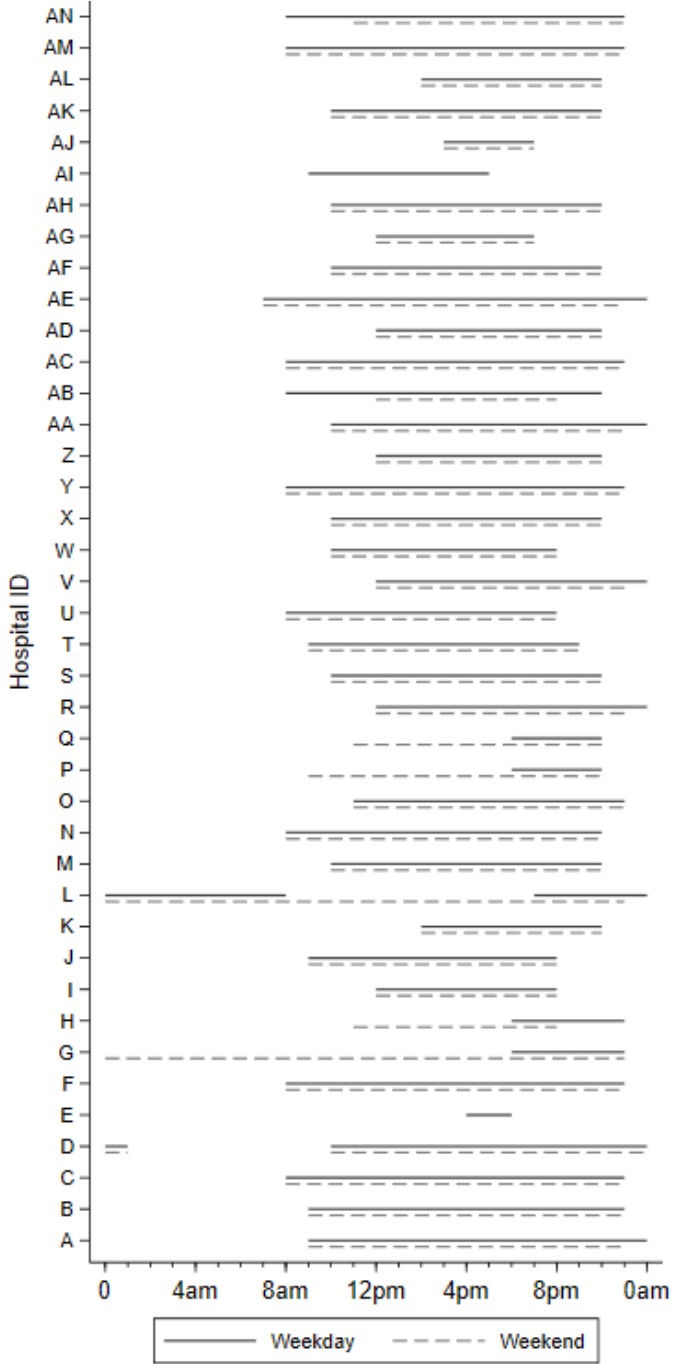

**Figure 1** General practitioner services in emergency department service availability by hour of day.

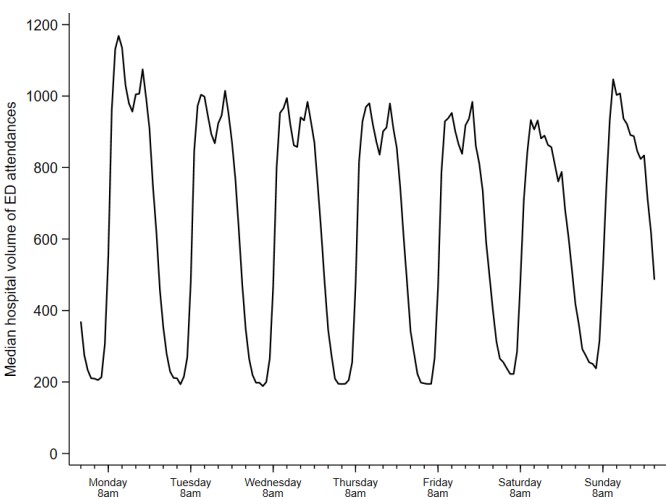

**Figure 2** Volume of attendances by hour of day. ED, emergency department.

on volume of attendances or on most performance indicators, including 4-hour waits, patients leaving without being treated or hospital admissions. We did identify a small, statistically significant reduction in unplanned reattendances within 7 days, equivalent to 302 fewer reattendances per year for the average ED. This is less than one prevented reattendance per day for the average ED. The finding was not statistically significant after adjusting for multiple testing and was judged to be of negligible clinical significance.

### Strengths and weaknesses
Our analysis exploited differences in the implementation of GPED services across 40 hospitals in terms of time of operation to model robustly their impact on a range of key indicators, accounting for a wide range of covariates. Using information from targeted surveys, we were also able to perform a subgroup analysis for three different GPED models, strengthening the evidence base to inform decisions about implementing GPED services. The use of

parallel GPED services and not for off-site GP services. None of these findings reach statistical significance when adjusted for multiple testing.

## DISCUSSION
### Statement of principal findings
Many hospitals in England have implemented GPED services with the aim of treating non-urgent patients more appropriately, thereby improving efficiency and patient experience. Using attendance data for 2018/2019, we found no statistically significant effect of GPED services

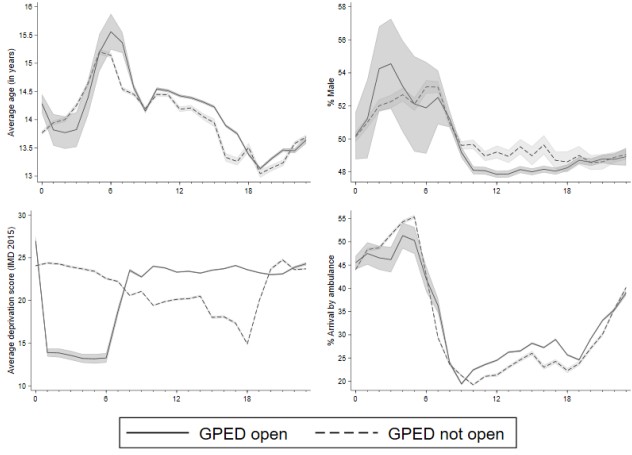

**Figure 3** Patient characteristics by hour of day (mean, 95% CI). GPED, general practitioner services in emergency department.

**Table 2** Impact of GPED service availability on outcomes, overall and by GPED model

| Outcome/GPED model | OR/IRR | 95% CI (unadjusted) | 95% CI (adjusted for multiple testing) |
|---|---|---|---|
| Wait over 4 hours (n=4 417 155 attendances) | | | |
| Overall | 1.012 | 0.927 to 1.105 | 0.870 to 1.153 |
| Integrated | 0.906 | 0.793 to 1.034 | 0.694 to 1.118 |
| Parallel | 0.997 | 0.863 to 1.132 | 0.783 to 1.211 |
| Off-site | 1.077 | 0.972 to 1.182 | 0.910 to 1.244 |
| Unplanned reattendance within 7 days (n=4 198 299 attendances) | | | |
| Overall | 0.968 | 0.948 to 0.989 | 0.935 to 1.001 |
| Integrated | 0.930 | 0.891 to 0.973 | 0.860 to 1.000 |
| Parallel | 0.972 | 0.945 to 0.998 | 0.929 to 1.014 |
| Off-site | 0.980 | 0.950 to 1.009 | 0.933 to 1.027 |
| Untreated (n=3 793 246 attendances) | | | |
| Overall | 0.913 | 0.787 to 1.060 | 0.696 to 1.130 |
| Integrated | 0.976 | 0.837 to 1.142 | 0.728 to 1.224 |
| Parallel | 0.926 | 0.730 to 1.124 | 0.611 to 1.241 |
| Off-site | 0.787 | 0.620 to 0.955 | 0.520 to 1.055 |
| Non-urgent attendance (n=4 441 349 attendances) | | | |
| Overall | 1.038 | 0.944 to 1.142 | 0.881 to 1.196 |
| Integrated | 1.014 | 0.844 to 1.219 | 0.722 to 1.306 |
| Parallel | 1.092 | 0.999 to 1.187 | 0.942 to 1.242 |
| Off-site | 0.990 | 0.874 to 1.106 | 0.805 to 1.175 |
| Admission to ward (n=3 793 246 attendances) | | | |
| Overall | 1.029 | 0.957 to 1.106 | 0.910 to 1.148 |
| Integrated | 1.039 | 1.016 to 1.185 | 0.916 to 1.161 |
| Parallel | 1.002 | 0.886 to 1.106 | 0.807 to 1.198 |
| Off-site | 1.047 | 0.957 tot 1.171 | 0.895 to 1.199 |
| 30-day mortality (n=4 076 605 attendances) | | | |
| Overall | 1.014 | 0.990 to 1.038 | 0.975 to 1.052 |
| Integrated | 1.041 | 0.968 to 1.119 | 0.926 to 1.156 |
| Parallel | 1.017 | 0.972 to 1.061 | 0.947 to 1.087 |
| Off-site | 1.000 | 0.968 to 1.032 | 0.948 to 1.053 |
| Volume of attendances (n=342 940 hospital–hour-day combinations) | | | |
| Overall | 1.000 | 0.967 to 1.034 | 0.946 to 1.054 |
| Integrated | 0.983 | 0.914 to 1.053 | 0.873 to 1.094 |
| Parallel | 1.018 | 0.967 to 1.069 | 0.937 to 1.099 |
| Off-site | 0.986 | 0.938 to 1.034 | 0.910 to 1.062 |

All models controlled for hospital and hour-by-day of week fixed effects, month-of-year fixed effects, and, in the case of the attendance-level regressions, patient age (coded as 0, 1, 2, 3, 4, 5, 6–10, 11–15 … 91–95, 96+), sex, age–sex interactions, an indicator for arrival by ambulance and socioeconomic deprivation (IMD) quintiles. Observations are dropped if any of the dependant or independent variables are missing. See online supplemental appendix A for details.
GPED, general practitioner services in emergency department; IMD, Index of Multiple Deprivation; IRR, incidence rate ratio.

time of day on which to base our analysis avoids a number of potential confounding factors, such as general hospital crowding or lack of community support.

Our analysis was, however, constrained in a number of ways. Most importantly, it was not possible to identify from available data which staff members assessed and treated individual patients, so we could not separate patients treated by GPs from those treated by other ED staff to directly compare GP services to traditional models of care. We relied primarily on measures of general ED performance, such as attendances, patient flow and waiting times. Our approach assumed GPs were present during the working hours reported in the survey, but there is potential for misclassifying patient episodes as GPED/not GPED as streaming activity and GP availability may not have corresponded exactly to official GPED start and finish times, particularly for patients arriving close to these times or undergoing extended waits. The start and finish times of GPED shifts may also have coincided with the shifts of other health professionals, such as advanced clinical practitioners, which have also been introduced to some EDs. Our design relies on no other major differences in service provision occurring at the same time as GPED shift hours. Our analysis also assumes that GPED was always operational during designated hours and therefore does not take account of physician absence. There were differences in observed outcomes between hospitals included in our analysis and those excluded from analysis (table 1). There were also substantial differences between the socioeconomic deprivation profile of patients attending hospitals with and without GPED services in the early hours of the day (figure 3). Finally, we were not able to assess the degree of 'role substitution', that is, whether GPs were an addition to normal services or whether they replaced another member of staff.

### Principal findings in the context of other studies

We found that patients attending EDs during the hours of operation of a GPED service had outcomes similar to those of patients attending at other times, with the possible exception of a very small reduction in unplanned reattendances within 7 days. Previous evidence on the impact of such services is extremely limited, and no previous studies have examined interventions on the same scale. A 2018 Cochrane review found no clear evidence that deploying primary care physicians in EDs reduced treatment times, lengths of ED stay, safety, hospital admissions, resource use, cost-effectiveness or referrals.

Future research in this area could usefully explore the implementation of GPED services, using more observational and qualitative methods, and include other important outcome measures (eg, those reflecting patient experience) which could improve data collection in EDs. The cost implications of these services should also be investigated further. Employing GPs in EDs has opportunity costs in terms of both alternative use of funds for urgent and emergency care (eg, employing additional nurses, physician associates or emergency physicians) and loss of GPs from primary care settings. Prior to the COVID-19 pandemic, there was a trend of increasing demand for both ED and primary care services, and general practice is an area of potential workforce shortages, with a forecasted shortfall of 7000

GPs in the next 5 years.[22] This raises further questions about the appropriateness of recruiting GPs to work in hospital ED settings.

## CONCLUSIONS

We found no adverse effects on patient outcomes; neither did we find any evidence of the hypothesised benefits of placing GPs in emergency settings beyond a marginal reduction in reattendances that was not considered clinically significant. Given that GPED services are likely to incur significant additional costs and to take GPs from primary care services which are themselves under pressure, we found no evidence that they are a good use of scarce healthcare resources in the absence of further development of the model and substantial improvements in outcomes.

**Contributors** The study was conceived and designed by JRB, KB, TD and NG. JG and DL prepared the dataset. JG and NG analysed the data, with additional interpretation by all other authors. JG drafted the manuscript, which was critically reviewed and revised by all other authors. All authors approved the final submission. JG is the guarantor of the overall content of the work.

**Funding** This study was funded by the National Institute of Health Research Health Services and Delivery Research Programme (project numbers 15/145/04 and 15/145/06).

**Disclaimer** The views expressed are those of the authors and not necessarily those of the NIHR or the Department of Health and Social Care.

**Competing interests** None declared.

**Patient consent for publication** Not applicable.

**Ethics approval** This research study was approved by: East Midlands-Leicester South Research Ethics Committee (Ref: 17/EM/0312), University of Newcastle Ethics Committee (Ref: 14348/2016) and Health Research Authority (IRAS: 230848 and 218038).

**Provenance and peer review** Not commissioned; externally peer reviewed.

**Data availability statement** No data are available. The deidentified patient-level data used in this study, including information on mortality, were released by the data holders (NHS Digital, Office for National Statistics) under specific data sharing agreements and only for the purpose of this study. The data sharing agreements do not permit further sharing or publication of the data. Interested parties may seek to obtain data directly from the relevant data holders. HES data are copyright 2018-2019, reused with the permission of NHS Digital through Data Sharing Agreement NIC-84254-J2G1Q. The data about the hours a general practitioner services was operating in emergency departments was collected by the authors specifically for this project. The authors are not able to place the original data into the public domain.

**ORCID iDs**
James Gaughan http://orcid.org/0000-0002-8409-140X
Karen Bloor http://orcid.org/0000-0003-4852-9854

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
