## [Reviewer comments · BMJ Open]

ARTICLE DETAILS

TITLE (PROVISIONAL)	Does the presence of General Practitioners in Emergency Departments affect quality and safety in English NHS Hospitals? A retrospective observational study
AUTHORS	Gaughan, James; Liu, Dan; Gutacker, Nils; Bloor, Karen; Doran, Tim; Bengner, Jonathan

VERSION 1 – REVIEW

REVIEWER	Lim, David Western Sydney University - Campbelltown Campus, School of Health Sciences
REVIEW RETURNED	22-Sep-2021

GENERAL COMMENTS	Thank you for the opportunity to peer-review this manuscript. The subject matter is something close to my heart and what our team has been working on in Australia. We previously published the data from one jurisdiction and more recently completed a rapid review on the topic, findings concurred with this study - potential for perceived bias. As the authors had identified, a strength of this manuscript is the sample size. I have nothing else to add and looking forward to reading and promoting the manuscript when published.
--

REVIEWER	Kelly, Shona Sheffield Hallam University, Faculty of Health and Wellbeing
REVIEW RETURNED	27-Sep-2021

GENERAL COMMENTS	This is an interesting and timely piece of work given the current issues in Emergency Departments and the almost complete dearth of research. However, it needs considerable work before publishing. The most important issues are: - It is not written for an international audience- The data is not well enough described for someone who doesn't already know the HES data- The conclusions and abstract overstate the confidence one could have in the findings given the nature of the data- There is plenty of unused word count that needs to be used to improve this paper Please note that I work with the HES data routinely as a research consultant for the NHS and spent 14 years working with the University of British Columbia Centre for Health Services and Policy Research which has been doing this kind of research since the late 1980's. I will now describe in more detail specific problems which need to be addressed
--

	Introduction  - is very limited and cursory. You could at least state that no other non-UK research could be found. Cochranes are severely constrained by their methodology. - It is clear from paragraph 2 that you were originally interested in safety and cost-effectiveness but you didn't address the former at all. Methods  - The HES data set is insufficiently described. What fields are used? How are they coded or recoded? An example is the explanation that outcomes 3 and 6 could not be calculated – what are the other options in that field? What are the age bands? – this matters because inappropriate ED use is often in patients classified in an extremely broad middle-age category that includes parents with young children, the sandwich generation, and students behaving stupidly. - It is really not clear which time points during the day were included. Improve the first paragraph in “Statistical analysis” - What is the evidence for validity of the 4 criteria used to determine “unnecessary attendance”? Also, what proportion of patients had sufficient data to meet this requirement (your reporting of 0 is inaccurate). - Elsewhere in the paper you describe small differences between the two groups of Eds for the outcome measures. Admission to ward is not a small difference. - How was IMD used? It is not normally used as a continuous measure. And it is better labelled as a measure on accumulated social disadvantage/advantage as it has multiple domains - Statistical analysis, 4th paragraph – What proportion of the ED visits were included in analyses? - State that OR/IRR and 95%CI are to 3 decimal places because of the high number of ED visits Results  - Figure 3 does not support your statement that IMD is similar between the two groups of Eds. I would label the differences in the figure as drastic. - Table 2 should stand alone without having to read the text. Insert a footer to state what variables were included in the fully adjusted model. Use longer subsection titles – e.g., “unplanned re-attendance within 30 days”, etc. - What are the n's for each analysis in Table 2? Discussion  - You need to acknowledge that the new ED-ACP roles overlap with the introduction of GPs and this new role is likely to have some overlap with GP patients. - A major limitation is that you weren't able to account for significant other factors such as: lack of beds, lack of community places to discharge to, staff shortages, etc. - I agree that there is a definite need for further research but the most crucial is an old fashioned time and motion study coupled with better data collection in ED. WRT the latter, it is telling that the 4 hour wait is almost complete as it is mandatory. Conclusions  - Your evidence isn't good enough to make this strong statement.
--	--

VERSION 1 – AUTHOR RESPONSE

Comment	Response
Reviewer: 1 Dr. David Lim, Western Sydney University - Campbelltown Campus, Flinders University	
Thank you for the opportunity to peer-review this manuscript. The subject matter is something close to my heart and what our team has been working on in Australia. We previously published the data from one jurisdiction and more recently completed a rapid review on the topic, findings concurred with this study - potential for perceived bias. As the authors had identified, a strength of this manuscript is the sample size. I have nothing else to add and looking forward to reading and promoting the manuscript when published.	Thank you for your comments.
Reviewer: 2 Prof. Shona Kelly, Sheffield Hallam University	
This is an interesting and timely piece of work given the current issues in Emergency Departments and the almost complete dearth of research. However, it needs considerable work before publishing. The most important issues are:	Thank you for your comments. We set out our responses below.
- It is not written for an international audience	We have added a more internationally focused introduction, and clarified any NHS-specific terms throughout.
- The data is not well enough described for someone who doesn't already know the HES data	We have expanded the description of HES data, see below.

- The conclusions and abstract overstate the confidence one could have in the findings given the nature of the data	We have amended the conclusion, see below
- There is plenty of unused word count that needs to be used to improve this paper Please note that I work with the HES data routinely as a research consultant for the NHS and spent 14 years working with the University of British Columbia Centre for Health Services and Policy Research which has been doing this kind of research since the late 1980's. I will now describe in more detail specific problems which need to be addressed	We have extended the introduction and data description particularly, and responded to your comments.
Introduction - is very limited and cursory. You could at least state that no other non-UK research could be found. Cochranes are severely constrained by their methodology.	As well as expanding the introduction for an international audience, we have highlighted the methodological constraints of a Cochrane review and added more detail from the two more inclusive reviews we mentioned very briefly in the previous version.
- It is clear from paragraph 2 that you were originally interested in safety and cost-effectiveness but you didn't address the former at all.	We have clarified the rationale of our study, which was to explore patient throughput and outcomes (some of these do, however, relate to patient safety). This study did not address cost-effectiveness questions.
Methods - The HES data set is insufficiently described. What fields are used? How are they coded or recoded? An example is the explanation that outcomes 3 and 6 could not be calculated – what are the other options in that field?	We have expanded paragraph 2 of the Study Design and Data Sources section, to include a general overview of the HES A&E dataset. In addition, the new Appendix A has been added to the paper, detailing the variables used to define the outcome variables analysed. Within the main text, the sentence "Information about the specific variables and variable values used to identify each outcome, along with reasons for missingness in each outcome measure, are presented in Appendix A." has been added to the section "Outcome Measures"

What are the age bands? – this matters because inappropriate ED use is often in patients classified in an extremely broad middle-age category that includes parents with young children, the sandwich generation, and students behaving stupidly.	Within the Data section we now note that the age field in HES is recorded in years. Then, in describing our regression model specifications, we state the specific age bands used for adjustment. We generally use 5-year age bands other than for children up to age of 5, where we use actual age, and for patients aged 96 or older, which are a single group.
- It is really not clear which time points during the day were included. Improve the first paragraph in “Statistical analysis”	We have expanded the first paragraph of Statistical Analysis to clarify this point. We now state explicitly that all hours of the day are used over the full study period and that we compare patients attending different EDs at the same hour of the day.
- What is the evidence for validity of the 4 criteria used to determine “unnecessary attendance”? Also, what proportion of patients had sufficient data to meet this requirement (your reporting of 0 is inaccurate).	The definition of ‘unnecessary attendance’ mirrors the definition used by NHS Digital and NHS England/Improvement to characterise ED utilisation, which itself is based on research conducted by Prof. Suzanne Mason and colleagues in Sheffield (Mason et al. 2017, O’Keeffe et al. 2018). We originally used the term ‘unnecessary’ as was previously used by NHS Digital (see also the URL to the NHS Digital website). We have now adopted the less contentious term ‘non-urgent’. We recognise that these terms are contested and have included relevant references. Mason S, O’Keeffe, Jacques R, Rimmer M, Ablard S. Perspectives on the reasons for Emergency Department attendances across Yorkshire and the Humber: Final Report. University of Sheffield. 2017. O’Keeffe C, Mason S, Jacques R, Nicholl J. Characterising non-urgent users of the emergency department (ED): a retrospective

	analysis of routine ED data. PLoS ONE. 2018; 13(2):e0192855 We have clarified in the section ‘Outcome measures’ that missingness cannot be ascertained for non-urgent attendances and 30-day mortality since both outcomes are defined according to the presence of some recorded data. If these data are not present, this may indicate that the outcome had not occurred – as we assume in our analysis – or that the outcome had occurred, but the data were not recorded to indicate so. We now state in Table 1 that the proportion of ‘missing’ data is ‘unknown’ rather than ‘0’.
- Elsewhere in the paper you describe small differences between the two groups of Eds for the outcome measures. Admission to ward is not a small difference.	The mean figure given in Table 1 for admissions at analysed Trusts was incorrect. The value referred to the standard deviation, not the mean. Table 1 has been updated to reflect this correction. The results of our statistical analysis (Table 2) are not impacted. We have included a new Appendix A which provides details about the reasons for missingness in dependant variables. We have also changed the wording in the discussion and removed the word ‘small’.
- How was IMD used? It is not normally used as a continuous measure. And it is better labelled as a measured on accumulated social disadvantage/advantage as it has multiple domains	We introduce IMD into our model as a categorical variable measuring socio-economic deprivation at the area level based on patient place of residence. The specific categories are set out in the Statistical Analysis section.

- Statistical analysis, 4th paragraph – What proportion of the ED visits were included in analyses?	The paragraph has been clarified. All attendances to an analysed ED in the study period is analysed. However, patients attending an ED at a time when all EDs are or are not operating a GPED service do not contribute specifically to the estimation of the impact of GPED. This includes 5.8% of the full sample of attendances.
- State that OR/IRR and 95%CI are to 3 decimal places because of the high number of ED visits	This has now been added to the section ‘Statistical analysis’
Results - Figure 3 does not support your statement that IMD is similar between the two groups of Eds. I would label the differences in the figure as drastic.	We have rephrased the Results section to point out this discrepancy more clearly, and have added a sentence to the discussion to reiterate this finding.
- Table 2 should stand alone without having to read the text. Insert a footer to state what variables were included in the fully adjusted model. Use longer subsection titles – e.g., “unplanned re-attendance within 30 days”, etc.	Table 2 has been updated with the proposed note, dependant variable names expanded and values of N included.
- What are the n’s for each analysis in Table 2?	See response to point above
Discussion - You need to acknowledge that the new ED-ACP roles overlap with the introduction of GPs and this new role is likely to have some overlap with GP patients.	As our design relies on time of day that GPs are present, rather than whether or not a service exists, the possibility of an ACP service being introduced at the same time (week, months, year) does not threaten the validity of our findings. We have, however, acknowledged the possibility of ACP and GPED timings coinciding.
- A major limitation is that you weren’t able to account for significant other factors such as: lack of beds, lack of community places to discharge to, staff shortages, etc.	We did not account for these factors explicitly but our analysis, based on the time of day that GPs are present in the ED (rather than the introduction of the service per se), minimises the potential confounding of some of these factors. We do accept, and have acknowledged, that our design relies on no

	other major service changes occurring alongside the GPED service.
- I agree that there is a definite need for further research but the most crucial is an old fashioned time and motion study coupled with better data collection in ED. WRT the latter, it is telling that the 4 hour wait is almost complete as it is mandatory.	We agree and have included both these points.
Conclusions - Your evidence isn't good enough to make this strong statement.	We have removed the first sentence of the conclusion, which we think was the strongest statement.

VERSION 2 – REVIEW

REVIEWER	Kelly, Shona Sheffield Hallam University, Faculty of Health and Wellbeing
REVIEW RETURNED	13-Jan-2022
GENERAL COMMENTS	I still feel that the multiplicity of care provision models will have increased the variance to the point that effects are difficult to see. Short of moving beyond regression you don't have a lot of options.